# Periodontitis and COVID-19: Immunological Characteristics, Related Pathways, and Association

**DOI:** 10.3390/ijms24033012

**Published:** 2023-02-03

**Authors:** Manlin Qi, Wenyue Sun, Kun Wang, Wen Li, Jinying Lin, Jing Gong, Lin Wang

**Affiliations:** Department of Oral Implantology, School and Hospital of Stomatology, Jilin University, Changchun 130021, China

**Keywords:** periodontitis, COVID-19, SARS-CoV-2, immunological mechanism, association

## Abstract

Both periodontitis and Coronavirus disease 2019 (COVID-19) pose grave threats to public health and social order, endanger human life, and place a significant financial strain on the global healthcare system. Since the COVID-19 pandemic, mounting research has revealed a link between COVID-19 and periodontitis. It is critical to comprehend the immunological mechanisms of the two illnesses as well as their immunological interaction. Much evidence showed that there are many similar inflammatory pathways between periodontitis and COVID-19, such as NF-κB pathway, NLRP3/IL-1β pathway, and IL-6 signaling pathway. Common risk factors such as gender, lifestyle, and comorbidities contribute to the severity of both diseases. Revealing the internal relationship between the two diseases is conducive to the treatment of the two diseases in an emergency period. It is also critical to maintain good oral hygiene and a positive attitude during treatment. This review covers four main areas: immunological mechanisms, common risk factors, evidence of the association between the two diseases, and possible interventions and potential targets. These will provide potential ideas for drug development and clinical treatment of the two diseases.

## 1. Introduction

Periodontitis is a chronic multifactorial inflammatory disease characterized by the gradual deterioration of dental supporting tissue and related to dysbiosis of plaque biofilms [1]. Severe periodontitis is considered the sixth most prevalent illness, affecting 11.2% of people worldwide [2]. Patients with periodontal disease have been troubled for a long time by low masticatory efficiency and phonetic changes, along with psychological disorders and quality of life seriously declined. Although periodontitis is not a fatal disease, it can harm human health. Epidemiologically, periodontitis increases the risk of most chronic noncommunicable illnesses, including cancer, coronary heart disease, and many other systemic disorders [3]. Therefore, periodontitis has become a severe public health and social problem.

Coronavirus disease 2019 (COVID-19) is an ongoing respiratory illness outbreak brought on by the coronavirus that causes severe acute respiratory syndrome (SARS-CoV-2) [4]. The World Health Organization (WHO) originally identified SARS-CoV-2 in December 2019 and afterwards referred to it as COVID-19 [5]. Over 656 million confirmed cases and 6.6 million fatalities have been documented globally to date [6]. The outbreak of COVID-19 seriously threatens global human health and poses a severe challenge to the public health, research, and medical community. Symptoms of COVID-19 patients demonstrate distinctive differences depending on the severity of the disease. Among them, severe COVID-19 patients show extensive clinical manifestations including acute respiratory distress syndrome, cytokine release syndrome (CRS), multiorgan failure, and even death [7]. In severe and critical COVID-19 patients, studies have shown that SARS-CoV-2 disrupts the normal immunological response, resulting in immune system dysfunction and uncontrolled inflammatory reactions [8].

Both periodontitis and COVID-19 severely impair human health and life. Moreover, increasing evidence demonstrates a correlation between periodontitis and COVID-19. Poor oral hygiene, as well as an increase in the incidence and severity of periodontitis, may exacerbate SARS-CoV-2 infection [9]. Moreover, periodontitis increased the risk of complications from COVID-19 and indirectly influenced COVID-19 outcomes by affecting systemic complications [10,11]. Although some issues of the association between periodontitis and COVID-19 were reviewed previously [12,13], the potential association between the immunopathogenesis of two diseases is pending further discussion. Treatments of killing two birds with one stone remain to be found and explained. In this review, we first summarize similar immunopathogenesis and signal pathways of periodontitis and COVID-19. Then, immunological evidence of an association between the two diseases is given. Finally, the association between periodontitis and COVID-19 is explored. On the one hand, this review provides readers with a quick overview of the disease profile and research status of periodontitis and COVID-19. On the other hand, it facilitates researchers identifying potential associations between the two diseases and doing further research in the next stage. In addition, this review explains in detail the immunological association between these two diseases, which has important implications for drug selection and clinical therapy.

## 2. Immunological Characteristics

### 2.1. Immunological Mechanism of Periodontitis

The interaction between the host and the bacterium leads to periodontitis, and early dysbiosis is driven by the host’s reaction. If the bacterial biofilm is not removed or microflora dysbiosis is, it will lead to the persistence of chronic destructive inflammation [14]. On the one hand, keystone pathogens can manipulate their interaction with the host immune response to enhance their adaptability. On the other hand, they can promote inflammation and avoid immune-mediated killing [15]. The immunopathology of periodontitis involves both innate and adaptive immunity. Together with the complement system, neutrophils, antigen-presenting cells, T lymphocytes, and B lymphocytes create a complex interaction network that contributes to the immunological and inflammatory response of periodontal tissue [16].

Neutrophils are considered the first line of defense of the host. Oral bacteria and their metabolites can activate the innate immune cell, which is then recruited into the damaged periodontal tissue through blood circulation. Neutrophils with a decreased number, hypofunction, or hyperactivity can lead to the progression of periodontitis. Patients with hereditary neutrophil abnormalities, such as congenital neutropenia, exhibit severe periodontitis at an early stage [17]. Keystone pathogens massively multiply due to immune escape. Periodontitis causes dysregulation of neutrophils, resulting in excessive recruitment, hyperactivity, or malfunction [18]. These effects more precisely include enhanced neutrophils phagocytosis, the increased generation of reactive oxygen species (ROS), proteases, as well as neutrophil extracellular traps (NETs) [18,19]. ROS can cause lipid peroxidation, and protein and DNA damage. Meanwhile, ROS, as one of the damage-associated molecular patterns (DAMPs), activates Nod-like receptor pyrin domain-containing protein 3 (NLRP3) inflammasome and promotes the release of proinflammatory factors [20,21]. Along with this process, it is also manifested in reduced collagen synthesis, the formation of osteoclasts, and the promotion of connective tissue destruction and bone resorption [21]. Proteases, particularly matrix metalloproteinases (MMPs), play an important role in pathological inflammation and malignant tissues. Hyperactive MMPs can almost degrade extracellular matrix and basement membrane components leading to periodontal tissue destruction [22]. Among them, the concentrations of MMP-8 and MMP-9 increased significantly in patients with periodontal disease and those with advanced periodontal disease, which can be used as potential markers for periodontitis screening and diagnosis [23,24]. NETs have a chromosomal structure surrounded by a variety of proteins and particles, which are released by neutrophils to capture and kill bacteria [25]. However, during acute inflammation, excessive NETs in infection may harm the host by the release of histones [19]. It is confirmed that not all neutrophils are fully activated trafficking through periodontal tissues [26]. Oral neutrophils with pro-inflammatory phenotype showed high levels of NETs in patients with chronic periodontitis [26].

Cytokines and chemokines are important factors in the onset and progression of periodontitis. A cytokine is a small molecule protein that is responsible for cellular signal transduction and communication. It is a key regulator to promote and balance inflammatory responses. Representative proinflammatory cytokines in periodontitis include most members of the interleukin family (IL), tumor necrosis factor family (TNF), granulocyte-macrophage colony-stimulating factor (GM-CSF), and prostaglandin E2 (PGE2) [27,28,29]. Chemokines are involved in the pathophysiology of periodontitis. Chemokines can chemotacticize and attract inflammatory cells (such as monocytes, lymphocytes, dendritic cells, and macrophages), as well as stimulate osteoclast migration and activation. Over-recruitment, on the other hand, can result in dysregulation of the inflammatory response and disturbance of tissue homeostasis [30]. Under the circumstances of inflammation and tissue damage, other significant pro-inflammatory mediators, such as complement, can be further activated, creating a vicious cycle that accelerates the development of illness [31].

### 2.2. Immunological Mechanism of COVID-19

COVID-19 is the result of host interactions with SARS-CoV-2. Direct damage from the virus, uncontrolled inflammatory responses, and virus-induced immune abnormalities lead to local or systemic damage. Currently, it is thought that the immunological etiology of COVID-19 may involve lymphopenia, neutrophilia, dysregulation of monocytes and macrophages, reduced or delayed type I interferon (IFN-I) response, antibody-dependent enhancement, as well as cytokine storm (CS) [32]. It suggests that during COVID-19, innate immunity plays a positive role, whereas adaptive immunity is functionally depleted.

SARS-CoV-2 infects host cells via angiotensin-converting enzyme 2 (ACE2) receptors and serine protease (TMPRSS2) [33]. The IFN I response program is induced by the recognition of pattern recognition receptors (PRRs) through Toll-like receptors (TLRs) or viral infection sensors [34]. It is currently believed that SARS-CoV-2 can inhibit and delay the induction of IFN I or inhibit the response of IFN I in infected cells [35,36]. SARS-CoV-2 can evade IFN I inducibility by inhibiting transcription and translation of host cells and can also target and control the induction of IFN I host proteins [37]. On the other hand, TLR3 responses can trigger the transcription of NLRP3 genes, which together with DAMPs produced by other cells in response to viruses, such as ROS, calcium ion release, and protein aggregation, are beneficial to activate the NLRP3 inflammasome [38]. Hyperactivation of neutrophils releases NETs and generates excess ROS in an oxidative burst, which can migrate to the local area through the blood system and cause a systemic inflammatory response [39]. The NLRP3 inflammasome is activated, causing CS and pyroptosis, which may operate as a COVID-19 therapeutic target and a biomarker of the severity of the disease [40,41].

Severe COVID-19 is distinguished by an excessive host immune response and significantly elevated levels of a diverse variety of pro-inflammatory cytokines and chemokines, known as CS [42,43,44]. Inflammatory cytokines and chemokines released after SARS-CoV-2 activate local innate immune cells, which subsequently recruit more innate immune cells and stimulate peripheral adaptive immune cells to create long-lasting inflammatory cytokines [32]. One longitudinal analysis study demonstrates immune malfunction in severe COVID-19 [45]. Moderate and severe patients have a core COVID-19 signature, that is, there are shared inflammatory cytokines which correlated positively with each other. A strong correlation between viral load and IFNα, IFNγ, and TNF is observed [45]. The unique combination of TNFα and IFNγ causes inflammatory cell death, which can prolong CS [46]. Moreover, most inflammatory cytokines in severe patients were found similar to those in CRS patients. However, in the late stage of infection, core signature makers steadily decreased in moderate patients whereas maintained a high level in severe patients [45]. A further characteristic of severe COVID-19 is complement activation. Prolonged proinflammatory cytokine production and excessive complement activation might result in multiorgan shock, damage, or failure [47,48].

### 2.3. Similar Pathways of Immune Response in Periodontitis and COVID-19

When bacteria or viruses invade tissues, the host experience innate and adaptive immunological responses. Furthermore, microbes can manipulate their interaction with the host immune response to escape innate immunity. There is increasing evidence that there is a strong inflammatory response in both periodontitis and COVID-19, and many of the same or similar immune response pathways exist in the two diseases.

#### 2.3.1. NF-κB Pathway

NF-κB signaling regulates innate and adaptive immunity. NF-κB is a critical mediator of the inflammatory response in the canonical route, causing the release of different pro-inflammatory proteins and participating in the control of the NLRP3 inflammasome [49]. In periodontitis, lipopolysaccharide from *Porphyromonas gingivalis*, a key pathogen, activates NF-κB through the TLR4/MyD88/NF-κB pathway, and causes nuclear translocation of NF-κB, induces the upregulation of MyD88 and NF-κB gene expression [50]. As a key transcription factor that regulates M1-type macrophage activation, NF-κB can promote the expression of pro-inflammatory factors [51]. Dysregulated NF-κB activation can lead to abnormal T cell activation such as T helper 1 (T_H_1) and T helper 17 (T_H_17) cells, which are implicated in autoimmune and inflammatory responses [49]. The recruitment of activated T_H_1 cells and pathogenic T_H_17 cells promote the progress of periodontitis [52,53]. During SARS-CoV-2 epidemics, various viral proteins were found to have the potential to hyperactivity NF-κB transcription. According to recent studies, the SARS-CoV-2 E and S protein are powerful PAMPs that can be detected by TLR2 or TLR4 and activate the NF-κB pathway, causing innate immune and generating inflammatory mediators [54,55]. RNA or nucleocapsid protein promote hyperactivity NF-κB transcription and inflammatory responses [56]. SARS-CoV-2 can over-activate NF-κB pathway through multiple pathways, including TLRs/MyD88/NF-κB and Angiotensin II type 1 receptor/NF-κB [55,57]. Excessive NF-κB activation induces a wide spectrum of pro-inflammatory cytokine as well as chemokine generation, leading to inflammatory cell activation and infiltration. It can also result in the production of numerous acute-phase proteins and adhesion molecules, causing vascular leak syndrome. All these eventually resulted in pneumonia and pulmonary edema [58] (Figure 1). Therefore, MyD88 and NF-κB are two important therapeutic targets.

#### 2.3.2. NLRP3/IL-1β Pathway

Numerous inflammatory and autoimmune illnesses are considered to include the NLRP3 inflammasome. Bacteria or viruses are recognized by PRRs after entering target cells. The recognition of DAMPs or pathogen-associated molecular patterns (PAMPs) causes NF-κB signaling to be activated. Nuclear NF-κB induces the transcriptional expression of NLRP3 and pro-IL-1β [59]. NLRP3 inflammasome activation causes caspase-1-dependent cleavage and release of the proinflammatory cytokines IL-1β and IL-18 and induces Gasdermin D-mediated pyroptosis, inflammation, and coagulation disorders [34]. Inflammatory cytokines could promote the development of systemic low-grade inflammation, and abnormal NLRP3 activation might fuel chronic inflammatory conditions, influencing the pathophysiology of illnesses associated with inflammation [60] (Figure 2).

Patients with periodontitis had considerably higher levels of NLRP3 in both their blood and saliva [61]. NLRP3 inflammasome-related proteins such as NLRP3, ASC, and IL-1β may act as potential biomarkers of periodontal clinical status. When compared to healthy controls, patients with chronic periodontitis had salivary levels of NLRP3, ASC, and IL-1β that were 2.94 times higher, 1.70 times higher, and 1.53 times higher, respectively [62]. NLRP3 regulates alveolar bone loss by promoting osteoclast differentiation [59]. Targeting the NLRP3 inflammasome is a choice for controlling age-related alveolar bone resorption [63]. Activation of dysregulated inflammasomes may lead to uncontrolled release of IL-1β. The severity of COVID-19 patients is correlated with the activation of the NLRP3 inflammasome [41]. According to postmortem findings, tissues and peripheral blood mononuclear cells from COVID-19-infected individuals who died still had active NLRP3 inflammasomes [41]. In severe and fatal COVID-19 cases, the inflammasome is the key to inducing large-scale inflammation [41,45]. Activation of the inflammasome is greatly amplified by positive feedback regulation, leading to uncontrolled hyperactivation and CS [64]. At the same time, positive feedback leading to the recruitment of more immune cells and a cascade of feed-forward tissue damage can synergistically exacerbate lung injury [64].

IL-1β is a key pro-inflammatory factor that can exacerbate acute and chronic inflammatory diseases, and tissue damage. It is mainly secreted by immune cells such as monocytes or macrophages in response to PAMPs or DAMPs [20]. IL-1β is involved in many key pathophysiological processes in vitro and in vivo including metabolism, physiologic response, hematologic response, inflammation, and immune response [65]. During these processes, functions of IL-1β involve immune cell activation, as well as the regulation of cytokines. In terms of clinical studies utilizing IL-1β neutralization, IL-1β is regarded as a gatekeeper of inflammation [65]. During the onset of periodontal disease, IL-1β stimulates endothelial cells and causes eosinophils to adhere, hence boosting the inflammatory response [66]. IL-1β can promote the secretion of MMPs, IL-6, and other molecules, accelerate collagen degradation, promote osteoclastogenesis, and accelerate bone resorption [67]. A large amount of IL-1β can aggravate inflammatory responses and autoimmune diseases [66]. IL-1β and other cytokines related to CRS showed a positive correlation in severe COVID-19 patients [45]. IL-1β and TNF are involved in the process of increased vascular permeability, vascular leakage, and coagulation, which results in vascular leakage, pulmonary edema, and disseminated intravascular coagulation in severe COVID-19 patients [68].

#### 2.3.3. IL-6 Signaling Pathway

A fast membrane-to-nucleus signaling module is composed of the Janus kinase family (JAK) and transcription factors (TFs) from the signal transducer and activator of the transcription (STAT) family [69]. More than 50 different molecules use the JAK/STAT signaling pathway to regulate many aspects of the mammalian immune system, including hematopoiesis, immunological response, inflammation, apoptosis, and tissue repair [70]. The JAK/STAT pathway is composed of ligand-receptor complexes, JAKs, and STATs.

IL-6 family, as key modulators of innate immunity, acts directly on innate immune cells and indirectly by activating stromal tissue cells at sites of inflammation [71]. In addition, IL-6 regulates adaptive immunity by inducing the maturation of B cells and the expansion and activation of T cells [72]. There are three modes of IL-6 signaling including classical IL-6 receptor (IL-6R) signaling, IL-6 trans-signaling, and IL-6 cluster signaling [73,74]. The classical IL-6R signaling pathway occurs in restricted cells which express both IL-6 receptor α-chain (IL-6Rα) and gp130. In contrast, IL-6 trans-signaling can occur in all gp130-expressing cells and lead to pro-inflammatory responses. In the third model, T cells were trans-presented with IL-6 to stimulate pathogenic T_H_17 cells by dendritic cells (DCs) positive for the signaling regulator Sirpα [74,75]. The JAK-STAT pathway is primarily used to transduce intracellular signals by IL-6-related cytokine receptor complexes [71]. IL-6 has a significant clinical relationship with the occurrence of various diseases. The dysregulation of IL-6 expression can cause many diseases or lead to the exacerbation of the disease.

IL-6 plays an important role both in periodontitis and COVID-19. Salivary IL-6 was shown to be a reliable predictor of gingival health. Patients with periodontitis had considerably higher levels of IL-6 in their saliva and serum. Salivary IL-6 levels increased significantly with periodontitis severity and tooth loss [76]. Recent random effects meta-analyses demonstrated intermediate quality evidence that individuals with periodontitis had greater serum IL-6 levels than those without periodontitis among transplanted patients. (MD: 2.20 (95% CI: 1.00, 3.39)) [77]. Han et al. revealed that IL-6-induced hepcidin is an important mediator of periodontitis-related inflammatory anemia. Hepcidin expression is influenced by inflammation and is mediated by IL-6/JAK/STAT3 signaling [78]. Serum IL-6 levels are correlated with the stage of COVID-19 as well as respiratory failure [79]. IL-6 can be utilized as a predictor for rapid identification of COVID-19 individuals at risk of illness progression [80]. The expression of IL6R was shown to be higher in COVID-19 patients activated CD4^+^ T cells, naïve T cells, and DCs when compared to healthy controls [81]. IL-6 was considerably higher in non-survivors than in survivors throughout the duration of the clinical course and increased with disease progression [82]. Therefore, the IL-6 blockade in COVID-19 is of vital importance and a global scientific call is needed to target the IL-6 blockade [83].

Both the induction and maintenance of T_H_17 cells depend on IL-6 signaling [84]. Pathogenic T_H_17 cells can promote the progress of both periodontitis and COVID-19. A dysbiotic microbiome induced IL-6 and IL-23-dependent increase of T_H_17 cells in periodontitis. Pathogenic T_H_17 cells secrete large amounts of IL-17 leading to excessive recruitment of neutrophils and associated immunopathology resulting in alveolar bone resorption [53]. Conversely, patients with autosomal dominant hyper-immunoglobulin E syndrome with defective differentiation of T_H_17 cells were not susceptible to periodontal disease and had significantly reduced periodontal inflammation and bone loss [85]. As a result, IL-6-mediated proinflammatory activity can be inhibited by blocking IL-6R, and alveolar bone absorption and attachment loss supported by Th17 periodontal response regulation can be decreased [86]. In the peripheral blood of COVID-19 patients, the number of CD4^+^ T cells is significantly reduced. Conversely, CD4^+^ T cells are hyperactivated and manifested an increase of T_H_17 cells [87]. In severe COVID-19, pulmonary tissue-resident memory T_H_17 cells (T_RM_17 cells) are regarded as one potential orchestrator of hyperinflammation. After the virus was eliminated, T_RM_17 cells were discovered in the lungs. T_RM_17 cells have been linked to lung damage and disease severity through their interactions with lung macrophages and cytotoxic CD8^+^ T cells [88] (Figure 3).

## 3. Association between Periodontitis and COVID-19

On the one hand, COVID-19 may directly infect oral tissues and cause a range of oral clinical manifestations, such as necrotic periodontal disease. On the other hand, chronic inflammation brought on by conditions such as periodontitis may increase a patient’s susceptibility to COVID-19, which will aggravate the course of COVID-19 and cause systemic inflammation. This section focuses on the relationship between periodontitis and COVID-19.

### 3.1. Common Risk Factors

#### 3.1.1. Gender

The most prevalent risk factor for periodontal disease is the male sex, which considerably raises the risk of periodontal disease [89]. According to the data from the National Health and Nutrition Examination Survey (NHANES) 2009–2014, a total of 42.2% of US adults aged 30 and older suffered from periodontitis, distributed as 7.8% having severe periodontitis (SP) and 34.4% having non-severe periodontitis (NSP) [90]. Male patients with total periodontitis are more than female patients (50.2% versus 34.6%). More males than females had SP (11.5% versus 4.3%) and NSP (38.8% versus 30.2%) [90]. Higher severity of COVID-19 and higher COVID-19 mortality in aggregate were found to be associated with the male gender [91,92]. Risk factor differences in gender were caused by hormonal variations in inflammatory processes, variations in ACE2 and TMPRSS2 expression levels, and lifestyle choices including smoking [91]. In contrast, female patients seem more likely to suffer from long-term consequences of COVID-19 such as reduced exercise tolerance and reduced resilience, and post-COVID syndrome [93]. 

#### 3.1.2. Lifestyle

A systematic review showed that smoking increased the risk of periodontitis by 85% (risk ratios (RR):1.85, 95% confidence interval (CI):1.5, 2.2) [94]. According to the data from NHANES 2009–2014, compared with nonsmokers (4.9%) and former smokers (8.0%), current smokers had the highest rate (16.9%) of suffering from severe periodontitis [90]. A 6-year follow-up suggested the association between cumulative smoking exposure, quitting smoking, and the recurrence of periodontitis (RP) in periodontal maintenance therapy. The RP in nonsmoker, former smoker, and current smoker groups was 44.2%, 68.2%, and 80.0%, respectively. Pack-years of smoking had a significant dose-response relationship with the RP. Additionally, the risk of RP decreased dramatically with the number of years since quitting smoking rose [95]. In many meta-analyses, smoking has been found to have an impact on the severity of COVID-19 [96,97]. Current smokers (RR:1.98; CI:1.16–3.38; *p* = 0.012) or former smokers (RR:1.35; CI:1.19–1.53; *p* < 0.0001) had an elevated risk of severe or critical COVID-19 [97]. New research that used the UK Biobank to conduct observational and Mendelian randomization studies confirmed the link between smoking and the risk of COVID-19 and its severity [98]. Compared with nonsmokers, current smokers had higher risks of hospitalization as well as mortality. With the extension of smoking time, the risk of mortality is higher.

Chronic excessive alcohol intake lowers resistance to bacterial and viral infections [99]. Alcohol consumption is related to the increased possibility of suffering from periodontitis, especially severe periodontitis. Individuals who consumed ≥8 or 1- < 8 drinks per week had higher values in clinical indicators including probing depth and clinical attachment level, compared with those who consumed <1 drink per week on average [100]. The frequency of alcohol consumption may lead to alveolar bone loss even in periodontitis-free models [101]. During the COVID-19 pandemic, it is recommended to avoid alcohol consumption as much as possible by specialists from the WHO [102]. Numerous pathophysiological processes such as the reduction of T cells and Natural Killers cells, favoring a pro-inflammatory status suggest the link between bacterial and viral lung infections and alcoholism [103]. Meanwhile, excessive alcohol consumption could slow down leukocyte proliferation and differentiation, leading to immunoglobulin malfunction [103].

#### 3.1.3. Comorbidities

Several systemic disorders and conditions can influence the progression of periodontitis by influencing periodontal inflammation or affecting the periodontal attachment apparatus rather than biofilm-induced inflammation [104]. The former mainly refers to hereditary disorders, acquired immunodeficiency diseases, inflammatory diseases, and some common systemic disorders, while the latter refers mainly to neoplastic diseases [105]. COVID-19 relies heavily on the immunological system. Many COVID-19-related comorbidities will impair immune system function and hence directly alter COVID-19 responsiveness. Several metabolic and infectious comorbidities impact the severity of COVID-19 [105]. According to recent literatures, the comorbidities of two diseases were summarized in Figure 4 [104,105,106,107,108,109,110,111,112,113].

### 3.2. Impact of Periodontitis on COVID-19

#### 3.2.1. The Oral Cavity Is a Reservoir for SARS-CoV-2

ACE2 and TMPRSS are important proteases for SARS-CoV-2 to enter host cells. Patients with COVID-19 were discovered to have epithelial cells in their saliva that were ACE2 and TMPRSS2 positive [114]. The expression of ACE2 and TMPRSS2 is upregulated in periodontitis gingiva [115,116]. Moreover, the cluster of differentiation 147 (CD147) binding to the S-protein, is found overexpressed in the oral, subgingival-component epithelial cells in patients with periodontitis [117]. These conditions favor the virus’s entrance into the oral host cell. SARS-CoV-2 RNA was detected in dental calculus, supragingival, and subgingival plaque biofilms in severe COVID-19 patients [118,119]. SARS-CoV-2 found in gingival crevicular fluid, dental plaque, and periodontal tissue suggests the association between COVID-19 and periodontitis [119,120,121,122].

#### 3.2.2. Oral Microorganisms Are Significant Contributors to COVID-19

Periodontal bacteria facilitate SARS-CoV-2 infection by boosting ACE2 expression. Periodontal bacteria release protease, which can enhance SARS-CoV-2 infectivity [123]. Periodontal bacteria and viruses can enter the lower airway via inhalation or aspiration from oral cavities and nasal cavities, called aspiration pneumonia [124]. Studies showed that the levels of oral symbiotic bacteria and upper respiratory symbiotic bacteria are increased in the lungs of COVID-19 patients [125]. Proinflammatory cytokines are released from the lower respiratory tract as a result of periodontal germs, and this may contribute to COVID-19 [123]. The role of oral bacteria in promoting co-infections in COVID-19 is relevant. Bacteremia induced by oral bacteria can promote the blood dissemination of inflammatory mediators and lead to systemic inflammation [126].

#### 3.2.3. Association between Periodontitis and COVID-19 in Clinical Studies

Periodontal condition is related to lung function impairment. According to data from NHANES III, with increasing severity of periodontitis, the values related to obstructive and restrictive pulmonary functions were found to gradually decline (*p* < 0.001) [127]. Case-control research revealed that the SARS-CoV-2 infection may become more severe due to the increasing incidence and severity of periodontitis as well as the associated poor oral hygiene [9]. Prospective observational research found a link between periodontitis and severe COVID-19 outcomes in COVID-19 patients who were hospitalized. Increased COVID-19 and a greater likelihood of ICU admission, severe-critical symptoms, the requirement for assisted ventilation, and mortality are linked to unfavorable oral and periodontal diseases [10,128,129]. One of the important reasons is that periodontitis can induce low-grade systemic inflammation, which probably affects the progress of inflammatory comorbidities [130]. These inflammatory comorbidities, in turn, may cause an excessive inflammatory response of the immune system in susceptible patients following COVID-19 infection [131]. Therefore, periodontitis could exacerbate COVID-19 infection by causing inflammatory comorbidities.

### 3.3. Impact of COVID-19 on Periodontitis

#### 3.3.1. Effects of Virus

According to previous studies, human viruses are believed to cause periodontal infection, not a bystander of periodontal disease [132]. The viruses usually exist in the oral cavity, causing a strong inflammatory reaction, and causing damage to the immune response [133]. Active viruses can also cooperate with bacteria to become a more pathogenic microbial complex, leading to severe periodontitis [133,134]. Based on the common characteristics of the virus, it is hypothesized that SARS-CoV-2 may cause periodontal infection. COVID-19 patients showed a wide range of oral symptoms, including necrotic periodontal disease (NPD), etc. It is predicted that the prevalence of NPD could be a spontaneous rise, in conformity to the growth of COVID-19 confirmed cases. The pathogenesis of NPD may be related to bacterial co-infection in the oral cavity of COVID-19 patients [135]. This clinical manifestation is reminiscent of the oral manifestations of necrotizing ulcerative gingivitis (NUG) in HIV-infected individuals. Immunocompromised or immunosuppressed HIV patients showed more severe clinical manifestations of periodontal disease [109]. In COVID-19, innate immunosuppression resulted in uncontrolled SARS-CoV-2 replication, which is the basis of dysregulated inflammatory response [136]. Therefore, it is predicted that COVID-19 may aggravate periodontitis through immunosuppression.

#### 3.3.2. Stress during the COVID-19 Outbreak and Lockdown

COVID-19 caused indelible psychological distress. An overwhelming majority (88%) report anxiety about a loved one infected with COVID-19 and almost half (43%) reported previous mental health treatment [137]. It was reported that the prevalence of posttraumatic stress disorder in the Chinese population was 4.6% one month after the COVID-19 outbreak [138]. Furthermore, the COVID-19 lockdown resulted in significant rates of unfavorable mental health consequences in the Italian people, including post-traumatic stress symptoms (37%), sadness (17.3%), anxiety (20.8%), sleeplessness (7.3%), perceived stress (21.8%), and adjustment disorder (22.9%) [139]. Stress is a risk factor for many diseases including periodontitis. Acute psychosocial stress stimulates systemic inflammatory responses and chronic psychosocial stress is related to systemic low-grade inflammation [140]. Chronic stress can cause changes in hormone levels (including glucocorticoids and catecholamines), nervous system changes, delayed healing, and so on [141]. The increase of cortisol could inhibit vital functions of inflammatory cells, causing dysregulation of the immune system [142]. For example, the absence of neutrophils could cause dysregulated overproduction of IL-17, driving inflammatory bone loss [18]. Social stress influences individual behaviors including smoking, dieting, and drinking alcohol, which are risk factors for periodontitis [142]. Therefore, the psychological stress caused by COVID-19 may become a risk factor for periodontitis.

## 4. Possible Interventions and Potential Targets

Various shreds of evidence indicate that periodontitis is associated with COVID-19. Therefore, it is very important to provide good periodontal care for patients with both diseases. Cleaning of tooth calculus and plaque biofilm, which can prevent COVID-19 colonization and infection of sick periodontal tissue, is part of the basic periodontal therapy. Simultaneously, increasing periodontal health can minimize the systemic inflammatory state and systemic problems induced by bacteremia generated by periodontal bacteria, lowering the likelihood of infection, or worsening of COVID-19. Some risk factors, including smoking cessation and alcohol consumption management, can be addressed to slow the progression of the two illnesses. Furthermore, both disorders are exacerbated by psychological stress. As a result, during the COVID-19 epidemic, it is critical to provide prompt and effective psychological treatment to patients.

The main concerns in treating patients with severe COVID-19 are antiviral medication and immunological regulation. Nucleotide analogs (Remdesivir), protease inhibitors (Paxlovid), and neutralizing monoclonal antibodies (Sotrovimab), among others, are examples of direct-acting antiviral medications [143]. Immune control techniques include immunosuppression and intervention against cytokines and pathways, with immunosuppressant medications such as dexamethasone, IL6 inhibitors such as tocilizumab or sarilumab, and JAK inhibitors serving as prominent examples (baricitinib or tofacitinib) [143]. According to the current newest USA treatment guidelines, SARS-CoV-2 subvariants are likely to be resistant to some anti-SARS-CoV-2 monoclonal antibodies. Therefore, the corresponding strategies should be adjusted according to the actual epidemiological situation. Conversely, due to the similarity in immune signal pathways between periodontitis and COVID-19, COVID-19’s immunomodulation therapy can improve periodontitis to a certain extent. Notably, clinical follow-up of the patient should be emphasized, as this allows the doctor to keep track of the patient’s condition. If the condition deteriorates, the treatment strategy can be modified in time. This is also an important part of assessing the effectiveness of clinical treatment [144]. However, there is a lack of research on the clinical follow-up of patients with combined periodontitis and COVID-19, which clinical researchers should focus on in the next phase.

## 5. Conclusions

Periodontitis and COVID-19 are related, as shown by immunological characteristics, shared risk factors, and clinical studies, and interaction, not causality, is what led to this link. Many similar inflammatory pathways exist in both diseases, including NF-κB pathway, NLRP3/IL-1β pathway, and IL-6 signaling pathway, which can promote systemic inflammation and aggravating complications. The male gender, lifestyle habits such as smoking and alcohol consumption, as well as comorbidities, are risk factors for exacerbating both diseases. The oral cavity is a reservoir for SARS-CoV-2 and bacteria promote viral infection of periodontal tissues. Therefore, in addition to antiviral treatment, periodontal care and initial periodontal therapy need to be taken seriously. For clinicians and patients, timely follow-up of patients after treatment is an important part of the process. What is worth noting is the psychological counseling for patients during the COVID-19 pandemic. It is expected that relevant departments will specify corresponding guidelines to standardize treatment strategies for patients suffering from periodontitis and COVID-19.

## Figures and Tables

**Figure 1 ijms-24-03012-f001:**
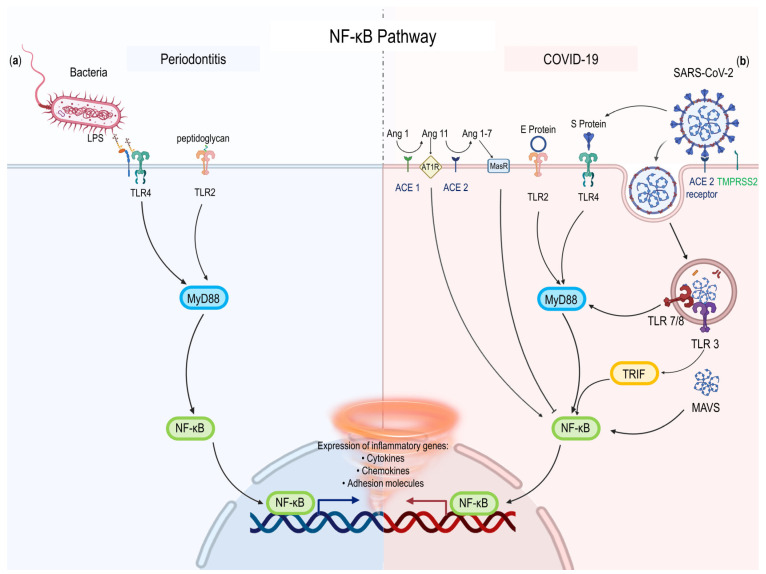
The NF-κB pathway in periodontitis and COVID-19. (**a**) Components of periodontitis pathogens (LPS and peptidoglycan) activate PRRs (TLR2 or TLR4). Then, these receptors activate MyD88/NF-κB pathway. (**b**) SARS-CoV-2 activates NF-κB via TLRs/MyD88/NF-κB and AT1R/NF-κB pathway. Then, the expression of inflammatory genes such as cytokines, chemokines, and adhesion molecules are upregulated. Legend. ACE2, angiotensin receptor 2; Ang, Angiotensin; AT1R, Angiotensin II type 1 receptor; LPS, lipopolysaccharide; MAVS, Mitochondrial Antiviral Signaling Protein; SARS-CoV-2, severe acute respiratory syndrome coronavirus-2; TLR, Toll-like receptor; TMPRSS2, Transmembrane Serine Protease 2.

**Figure 2 ijms-24-03012-f002:**
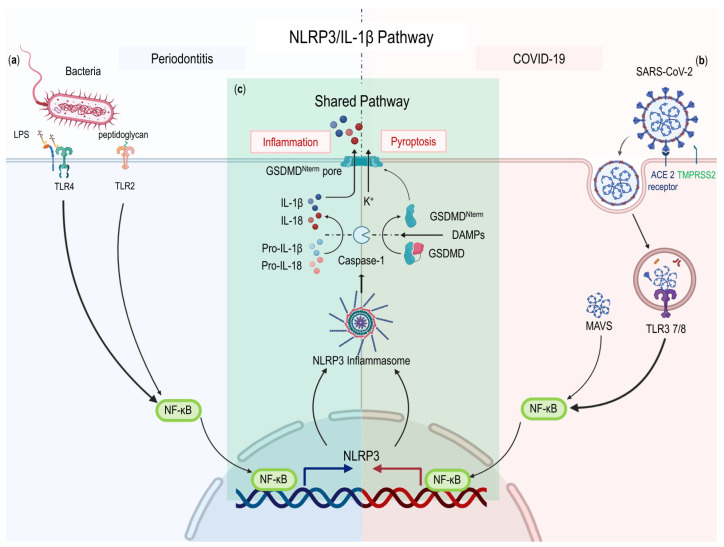
The NLRP3/IL-1β pathway in periodontitis and COVID-19. (**a**) Components of periodontitis pathogens (LPS and peptidoglycan) activate TLR4/NF-κB and TLR2/NF-κB pathway, respectively. (**b**) SARS-CoV-2 bind to ACE2 receptor or TMPRSS2 to entry cell and activate TLR3 or 7/8 and MAVS. Then, these receptors activate NF-κB signaling. (**c**) NLRP3/IL-1β pathway (Shared pathway). NF-κB triggered NLRP3 inflammasome. Caspase 1 is activated during inflammasome formation, and it then cleave pro-IL-1β, pro-IL-18, and GSDMD. Eventually, the release of IL-1β, IL-18, and GSDMD induce inflammation and pyroptosis. Legend. ACE2, angiotensin receptor 2; GSDMD, Gasdermin D; GSDMDNterm, GSDMD amino-terminal cell death domain; LPS, lipopolysaccharide; MAVS, Mitochondrial Antiviral Signaling Protein; SARS-CoV-2, severe acute respiratory syndrome coronavirus-2; TLR, Toll-like receptor; TMPRSS2, Transmembrane Serine Protease 2.

**Figure 3 ijms-24-03012-f003:**
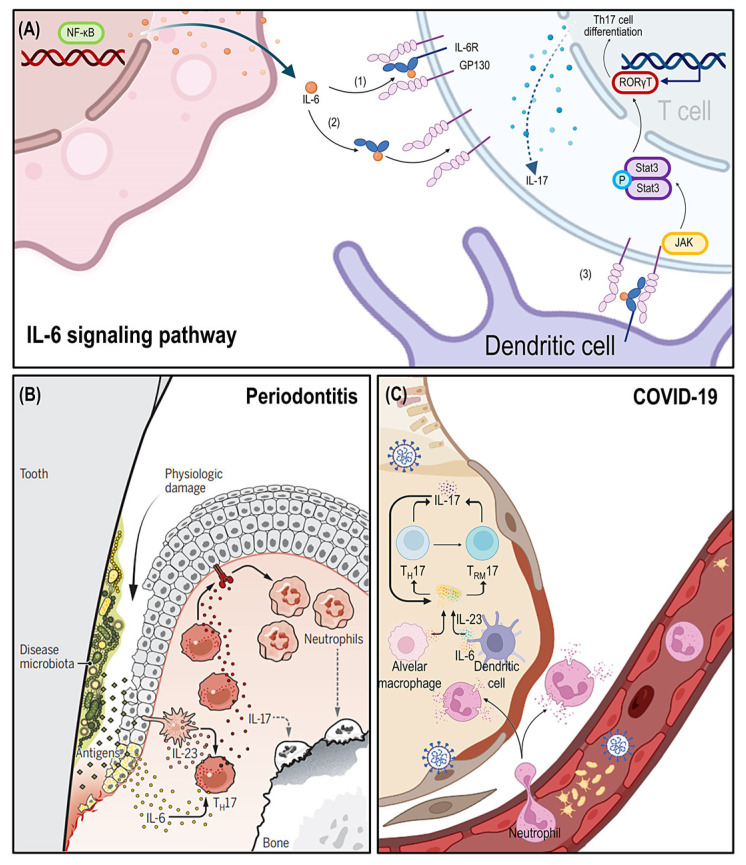
The IL-6 signaling pathway and microbe-induced T_H_17 cells in immunity of periodontitis and COVID-19. (**A**) The modes of IL-6 signaling are including (1) classical IL-6R signaling, (2) IL-6 trans-signaling, and (3) IL-6 cluster signaling. (**B**) Periodontal pathogens induce IL-6 and IL-23 dependent accumulation of pathological T_H_17 cells, which drive periodontal bone resorption through neutrophil over-recruitment and associated immunopathology. (Reproduced with permission from [53], AAAS, 2016.) (**C**) SARS-CoV-2 induce IL-6 and IL-23 dependent accumulation of pathological T_H_17 cells. Some T_H_17 cells persist in the lung by turning into T_RM_17 cells. Both T_H_17 cells and T_RM_17 cells respond to cytokines and secret IL-17, which thereby leads to hyperinflammation. Legend. GP130, Glycoprotein130; RORγT, retinoic-acid-receptor-related orphan nuclear receptor gamma; SARS-CoV-2, severe acute respiratory syndrome coronavirus-2; TH cells, T helper cells.

**Figure 4 ijms-24-03012-f004:**
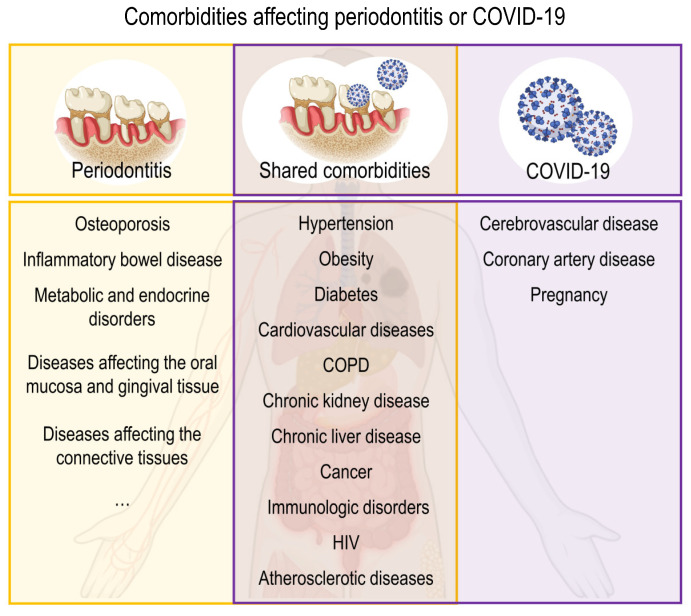
The comorbidities and shared comorbidities of periodontitis and COVID-19.

## Data Availability

Not applicable.

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
