# Peer review of "Periodontitis and COVID-19: Immunological Characteristics, Related Pathways, and Association"

_ijms, 2023, doi:10.3390/ijms24033012_

Round 1

Reviewer 1 Report

Dear authors,

Congratulations for work. The subject is of interest worldwide, as a proof the important number of articles on the same topic, some of them review articles, too (you can introduce also the reference below: Campisi, G., Bizzoca, M.E. & Lo Muzio, L. COVID-19 and periodontitis: reflecting on a possible association. Head Face Med 17, 16 (2021). https://doi.org/10.1186/s13005-021-00267-1).

The Abstract is rather short, even if the article is a review and you’ve tried to be concise.

In my opinion you can write more clearly the objectives of the article, at the end of the Introduction.

Regarding the Conclusions, I appreciate them as well written, in a concise manner, related to the structure of the article.

Author Response

Thank you very much for your excellent review comments and suggestions.

1.Congratulations for work. The subject is of interest worldwide, as a proof the important number of articles on the same topic, some of them review articles, too (you can introduce also the reference below: Campisi, G., Bizzoca, M.E. & Lo Muzio, L. COVID-19 and periodontitis: reflecting on a possible association. Head Face Med 17, 16 (2021). https://doi.org/10.1186/s13005-021-00267-1).

Thanks for your kind reminding. We read this article carefully. The statement “a direct link, through the ACEII and CD147 receptors used by the virus to infect the cells” mentioned in the article are not reviewed in this article. We think it is very necessary to add this to our original text. We added this in part Impact of periodontitis on COVID-19 The oral cavity is a reservoir for SARS-CoV-2. “Besides, the cluster of differentiation 147 (CD147) binding to the S-protein, is found overexpressed in the oral, subgingival-component epithelial cells in patients with periodontitis [117].”

  1. Campisi, G.; Bizzoca, M. E.; Lo Muzio, L., COVID-19 and periodontitis: reflecting on a possible association. Head Face Med 2021, 17, 16.

2.The Abstract is rather short, even if the article is a review and you’ve tried to be concise.

Yes, I agree. We revised the abstract and added details so that readers can quickly understand the main content of the whole article. The latest version is below.

“Both periodontitis and COVID-19 pose grave threats to public health and social order, endanger human life, and place a significant financial strain on the global healthcare system. Since the COVID-19 pandemic, mounting research has revealed a link between COVID-19 and periodontitis. It is critical to comprehend the immunological mechanisms of the two illnesses as well as their immunological interaction. Much evidence showed that there are many similar inflammatory pathways between periodontitis and COVID-19, such as including NF-κB pathway, NLRP3/IL-1β pathway, and IL-6 signaling pathway. Common risk factors such as gender, lifestyle and comorbidities contribute to the severity of both diseases. Revealing the internal relationship between the two diseases is conducive to the treatment of the two diseases in an emergency period. It is also critical to maintain good oral hygiene and a positive attitude during treatment. This review covers four main areas: immunological mechanisms, common risk factors, evidence of the association between the two diseases, and possible interventions and potential targets. These will provide potential ideas for drug development and clinical treatment of the two diseases.”

3.In my opinion you can write more clearly the objectives of the article, at the end of the Introduction.

Thanks. It is essential to give the purpose of writing this review. So, we revised the end of the Introduction. “On the one hand, this review provides readers with a quick overview of the disease profile and research status of periodontitis and COVID-19. On the other hand, it facilitates researchers to identify potential associations between the two diseases and do further research in the next stage. In addition, this review explains in detail the immunological association between these two diseases, which has important implications for drug selection and clinical therapy.”

4.Regarding the Conclusions, I appreciate them as well written, in a concise manner, related to the structure of the article.

Thanks. We rewrite the Conclusion following the structure of the article.

“Periodontitis and COVID-19 are related, as shown by immunological characteristics, shared risk factors, and clinical studies. Many similar inflammatory pathways exist in both diseases, including NF-κB pathway, NLRP3/IL-1β pathway, and IL-6 signaling pathway, which can promote systemic inflammation and aggravating complications. The male gender, lifestyle habits such as smoking and alcohol consumption, as well as comorbidities, are risk factors for exacerbating both diseases. The oral cavity is a reservoir for SARS-CoV-2 and bacteria promote viral infection of periodontal tissues. Therefore, in addition to antiviral treatment, periodontal care and initial periodontal therapy need to be taken seriously. For clinicians and patients, timely follow-up of patients after treatment is an important part of the process. What is worth noting is the psychological counseling for patients during the COVID-19 pandemic. It is expected that relevant departments will specify corresponding guidelines to standardize treatment strategies for patients suffering from periodontitis and COVID-19.”

Reviewer 2 Report

GIVEN THE HIGHLIGHTED CORRELATION BETWEEN COVID AND PERIODONTAL DISEASE, IT IS IMPORTANT TO FURTHER EMPHASISE THAT ORAL HYGIENE FOLLOW-UPS ARE CRUCIAL FOR THE DEFENCE OF BOTH DISEASES DESCRIBED

WE RECOMMEND THE FOLLOWING PUBLICATIONS ON BOTH HEALTHY AND IMPLANT-REHABILITATED PATIENTS

Introduction and discussion should be improved with the suggested references

AUTHORS SHOULD PLACE MORE EMPHASIS ON THE IMPORTANCE OF HYGIENE FOLLOW-UPS IN IMPLANT REHABILITATION CASES USING THE RECOMMENDED BIBLIOGRAPHIES

PMID: PubMed ID 34425659

PubMed ID 34425664

Materials and methods are well described and pertinent

CONCLUSION is correct and interesting.

Author Response

Thanks for your constructive comments. We read these studies carefully. As you mentioned, clinical follow-up needs to be paid attention, which is what the current research lacks. We added the content in part Possible interventions and potential targets. “Notably, clinical follow-up of the patient should be emphasized, as this allows the doctor to keep track of the patient's condition. If the condition deteriorates, the treatment strategy can be modified in time. This is also an important part of assessing the effectiveness of clinical treatment [144]. However, there is a lack of research on the clinical follow-up of patients with combined periodontitis and COVID-19, which clinical researchers should focus on in the next phase.”

  1. Cattoni, F.; Tete, G.; D'Orto, B.; Bergamaschi, A.; Polizzi, E.; Gastaldi, G., Comparison of hygiene levels in metal-ceramic and stratified zirconia in prosthetic rehabilitation on teeth and implants: a retrospective clinical study of a three-year follow-up. J Biol Regul Homeost Agents 2021, 35, 41-49.
